

Nitrogen isotope fractionation during gas-particle conversion of $NO_x$ to
$NO_3^-$ in the atmosphere – implications for isotope-based $NO_x$ source
apportionment
Yunhua Chang[1], Yanlin Zhang[1*], Chongguo Tian[2], Shichun Zhang[3], Xiaoyan Ma[4], Fang
Cao[1], Xiaoyan Liu[1], Wenqi Zhang[1], Thomas Kuhn[5], and Moritz F. Lehmann[5]
[1]Yale-NUIST Center on Atmospheric Environment, Nanjing University of Information
Science and Technology, Nanjing 10044, China
[2]Key Laboratory of Coastal Environmental Processes and Ecological Remediation,
Yantai Institute of Coastal Zone Research, Chinese Academy of Sciences, Yantai
264003, China
[3]Northeast Institute of Geography and Agroecology, Chinese Academy of Sciences,
4888 Shengbei Road, Changchun 130102, China
[4]Key Laboratory for Aerosol Cloud-Precipitation of China Meteorological
Administration, Earth System Modeling Center, Nanjing University of Information
Science and Technology, Nanjing 10044, China
[5]Aquatic and Isotope Biogeochemistry, Department of Environmental Sciences,
University of Basel, Basel 4056, Switzerland
[*] Corresponding author: Yanlin Zhang
E-mail address: dryanlinzhang@outlook.com




**Abstract**

Atmospheric fine-particle (PM$_{2.5}$) pollution is frequently associated with the formation

of particulate nitrate ($p$NO$_3^-$), the end product of the oxidation of NO$_x$ gases (=NO+NO$_2$)

in the upper troposphere. The application of stable nitrogen (N) (and oxygen) isotope

analyses of $p$NO$_3^-$ to constrain NO$_x$ source partitioning in the atmosphere requires the

knowledge of the isotope fractionation during the reactions leading to nitrate formation.

Here we determined the $\delta^{15}$N values of fresh $p$NO$_3^-$ ($\delta^{15}$N-$p$NO$_3^-$) in PM$_{2.5}$ at a rural site

in Northern China, where atmospheric $p$NO$_3^-$ can be attributed exclusively to biomass

burning. The observed $\delta^{15}$N-$p$NO$_3^-$ (12.17±1.55‰; n=8) was much higher than the N

isotopic source signature of NO$_x$ from biomass burning (1.04±4.13‰). The large

difference between $\delta^{15}$N-$p$NO$_3^-$ and $\delta^{15}$N-NO$_x$ ($\Delta(\delta^{15}$N)) can be reconciled by the net N

isotope effect ($\varepsilon_N$) associated with the gas-particle conversion from NO$_x$ to NO$_3^-$. For

the biomass-burning site, a mean $\varepsilon_N$ ($\approx \Delta(\delta^{15}$N)) of 10.99±0.74‰ was assessed through

a newly-developed computational quantum chemistry (CQC) module. $\varepsilon_N$ depends on

the relative importance of the two dominant N isotope exchange reactions involved

(NO$_2$ reaction with OH versus hydrolysis of dinitrogen pentoxide (N$_2$O$_5$) with H$_2$O),

and varies between regions, and on a diurnal basis. A second, slightly higher CQC-

based mean value for $\varepsilon_N$ (15.33±4.90‰) was estimated for an urban site with intense

traffic in Eastern China, and integrated in a Bayesian isotope mixing model to make

isotope-based source apportionment estimates for NO$_x$ at this site. Based on the $\delta^{15}$N

values (10.93±3.32‰, n=43) of ambient $p$NO$_3^-$ determined for the urban site, and

considering the location-specific estimate for $\varepsilon_N$, our results reveal that the relative

contribution of coal combustion and road traffic to urban NO$_x$ are 32±11% and 68±11%,

respectively. This finding agrees well with a regional bottom-up emission inventory of

NO$_x$. Moreover, the variation pattern of OH contribution to ambient $p$NO$_3^-$ formation

calculated by the CQC module is consistent with that simulated by the Weather

Research and Forecasting model coupled with Chemistry (WRF-Chem), further

confirming the robustness of our estimates. Our investigations also show that, without

the consideration of the N isotope effect during $p$NO$_3^-$ formation, the observed $\delta^{15}$N-





$p$NO$_3^-$ at the study site would erroneously imply that NO$_x$ is derived almost entirely
from coal combustion. Similarly, reanalysis of reported $\delta^{15}$N-NO$_3^-$ data throughout
China and its neighboring areas suggests that, nationwide, NO$_x$ emissions from coal
combustion may be substantively overestimated (by >30%) when the N isotope
fractionation during atmospheric $p$NO$_3^-$ formation is neglected.

## 1 Introduction

Nitrogen oxides (NO$_x$ = NO + NO$_2$) are among the most important molecules in
tropospheric chemistry. They are involved in the formation of secondary aerosols and
atmospheric oxidants, such as ozone (O$_3$) and hydroxyl radicals (OH), which controls
the self-cleansing capacity of the atmosphere (Galloway et al., 2003; Seinfeld and
Pandis, 2012; Solomon et al., 2007). The sources of NO$_x$ include both anthropogenic
and natural origins, with more than half of the global burden ($\sim$40 Tg N yr$^{-1}$) currently
attributed to fossil fuel burning (22.4-26.1 Tg N yr$^{-1}$) and the rest primarily derived
from nitrification/denitrification in soils (including wetlands; 8.9 ± 1.9 Tg N yr$^{-1}$),
biomass burning (5.8 ± 1.8 Tg N yr$^{-1}$), lightning (2-6 Tg N yr$^{-1}$), and oxidation of N$_2$O
in the stratosphere (0.1-0.6 Tg N yr$^{-1}$) (Jaegle et al., 2005; Richter et al., 2005; Lamsal
et al., 2011; Price et al., 1997; Yienger and Levy, 1995; Miyazaki et al., 2017; Duncan
et al., 2016; Anenberg et al., 2017; Levy et al., 1996). The main/ultimate sinks for
NO$_x$ in the troposphere are the oxidation to nitric acid (HNO$_{3(g)}$) and the formation of
aerosol-phase particulate nitrate ($p$NO$_3^-$) (Seinfeld and Pandis, 2012), the partitioning
of which may vary on diurnal and seasonal time scales (Morino et al., 2006).
Emissions of NO$_x$ occur mostly in the form of NO (Seinfeld and Pandis, 2012; Leighton,
1961). During daytime, transformation from NO to NO$_2$ is rapid (few minutes) and
proceeds in a photochemical steady state, controlled by the oxidation of NO by O$_3$ to
NO$_2$, and the photolysis of NO$_2$ back to NO (Leighton, 1961):
(R1)     $NO + O_3 \longrightarrow NO_2 + O_2$
(R2)     $NO_2 + h\nu \longrightarrow NO + O$





(R3)      $O + O_2 \xrightarrow{M} O_3$,
where M is any non-reactive species that can take up the energy released to stabilize
O. $NO_x$ oxidation to $HNO_3$ is governed by the following equations. During daytime:
(R4)      $NO_2 + OH \xrightarrow{M} HNO_3$,
and during nighttime:
(R5)      $NO_2 + O_3 \longrightarrow NO_3 + O_2$
(R6)      $NO_3 + NO_2 \xrightarrow{M} N_2O_5$
(R7)      $N_2O_5 + H_2O_{(surface)} \xrightarrow{aerosol} 2HNO_3$.
$HNO_3$ then reacts with gas-phase $NH_3$ to form ammonium nitrate ($NH_4NO_3$) aerosols.
If the ambient relative humidity (RH) is lower than the efflorescence relative humidity
(ERH) or crystallization relative humidity (CRH), solid-phase $NH_4NO_3(s)$ is formed
(Smith et al., 2012; Ling and Chan, 2007):
(R8a)     $NH_4NO_3 \rightleftharpoons HNO_3(g) + NH_3(g)$.
If ambient RH exceeds the ERH or CRH, $HNO_3$ and $NH_3$ dissolve into the aqueous
phase (aq) (Smith et al., 2012; Ling and Chan, 2007):
(R8b)     $HNO_3(g) + NH_3(g) \rightleftharpoons NO_3^-(aq) + NH_4^+(aq)$.
Whilst global $NO_x$ emissions are well constrained, individual source attribution and
their local or regional role in particulate nitrate formation are difficult to assess due to
the short lifetime of $NO_x$ (typically less than 24 hr), and the high degree of
spatiotemporal heterogeneity with regards to the ratio between gas-phase $HNO_3$ and
particulate $NO_3^-$ ($pNO_3^-$) (Duncan et al., 2016; Lu et al., 2015; Zong et al., 2017; Zhang
et al., 2003). Given the conservation of the nitrogen (N) atom between $NO_x$ sources and
sinks, the N isotopic composition of $pNO_3^-$ can be related to the different origins of the





emitted $NO_x$, and thus provides valuable information on the partitioning of the $NO_x$
sources. Such N isotope balance approach works best if the N isotopic composition of
various $NO_x$ sources display distinct $^{15}N/^{14}N$ ratios (reported as
$\delta^{15}N = \dfrac{\left(^{15}N/^{14}N\right)_{sample} - \left(^{15}N/^{14}N\right)_{N_2}}{\left(^{15}N/^{14}N\right)_{N_2}} \times 1000$ ). The $\delta^{15}$N-$NO_x$ of coal-fired power plant
(+10‰ to +25‰) (Felix et al., 2012; Heaton, 1990; Felix et al., 2013), vehicle (+3.7‰
to +5.7‰) (Heaton, 1990; Walters et al., 2015; Felix and Elliott, 2014; Felix et al., 2013;
Wojtal et al., 2016), and biomass burning (-7‰ to +12‰) emissions (Fibiger and
Hastings, 2016), for example, are generally higher than that of lightning (-0.5‰ to
+1.4‰) (Hoering, 1957) and biogenic soil (-48.9‰ to -19.9‰) emissions (Li and Wang,
2008; Felix and Elliott, 2014; Felix et al., 2013), allowing the use of isotope mixing
models to gain insight on the $NO_x$ source apportionment for gases, aerosols, as well as
the resulting nitrate deposition (-15‰ to +15‰) (Elliott et al., 2007; Zong et al., 2017;
Savarino et al., 2007; Morin et al., 2008; Elliott et al., 2009; Park et al., 2018; Altieri et
al., 2013; Gobel et al., 2013). In addition, because of mass-independent fractionation
during its formation (Thiemens, 1999; Thiemens and Heidenreich, 1983), ozone
possesses a strong isotope anomaly ($\Delta^{17}O \approx \delta^{17}O - 0.52*\delta^{18}O$), which is propagated into
the most short-lived oxygen-bearing species, including $NO_x$ and nitrate. Therefore, the
oxygen isotopic composition of nitrate ($\delta^{18}O$, $\Delta^{17}O$) can provide information on the
oxidants involved in the conversion of $NO_x$ to nitrate (Michalski et al., 2003; Geng et
al., 2017).
$\delta^{15}$N-based source apportionment of $NO_x$ requires knowledge of how kinetic and
equilibrium isotope fractionation may impact $\delta^{15}$N values during the conversion of $NO_x$
to nitrate (Freyer, 1978; Walters et al., 2016). If these isotope effects are considerable,
they may greatly limit the use of $\delta^{15}$N values of $p$NO$_3^-$ for $NO_x$ source partition (Walters
et al., 2016). Previous studies didn't take into account the potentially biasing effect of
N isotope fractionation, because they assumed that changes in the $\delta^{15}$N values during
the conversion of $NO_x$ to nitrate are minor (without detailed explanation) (Kendall et
al., 2007; Morin et al., 2008; Elliott et al., 2007) or relatively small (e.g., +3‰) (Felix



and Elliott, 2014; Freyer, 2017). However, a field study by Freyer et al. (1993) has
indicated that N isotope exchange may have a strong influence on the observed $\delta^{15}N$
values in atmospheric NO and $NO_2$, implying that isotope equilibrium fractionation
may play a significant role in shaping the $\delta^{15}N$ of $NO_y$ species (the family of oxidized
nitrogen molecules in the atmosphere, including $NO_x$, $NO_3$, $NO_3^-$, peroxyacetyl nitrate
etc.). The transformation of $NO_x$ to nitrate is a complex process that involves several
different reaction pathways (Walters et al., 2016). To date, few fractionation factors for
this conversion have been determined. Recently, Walters and Michalski (2015) and
Walters et al. (2016) used computational quantum chemistry methods to calculate N
isotope equilibrium fractionation factors for the exchange between major $NO_y$
molecules and confirmed theoretical predictions that $^{15}N$ isotopes enrich in the more
oxidized form of $NO_y$, and that the transformation of $NO_x$ to atmospheric nitrate ($HNO_3$,
$NO_3$ (aq), $NO_3$ (g)) continuously increases the $\delta^{15}N$ in the residual $NO_x$ pool.
As a consequence of its severe atmospheric particle pollution during the cold season,
China has made great efforts toward reducing $NO_x$ emissions from on-road traffic (e.g.,
improving emission standards, higher gasoline quality, vehicle travel restrictions) (Li
et al., 2017). Moreover, China has continuously implemented denitrogenation
technologies (e.g., selective catalytic reduction or SCR) in the coal-fired power plants
sector since the mid-2000s, and has been phasing out small inefficient units (Liu et al.,
2015). Monitoring and assessing the efficiency of such mitigation measures, and
optimizing policy efforts to further reduce $NO_x$ emissions, requires knowledge of the
vehicle- and power plant-emitted $NO_x$ to particulate nitrate in urban China (Ji et al.,
2015; Fu et al., 2013; Zong et al., 2017). In this study, the chemical components of
ambient fine particles ($PM_{2.5}$) were quantified, and the isotopic composition of
particulate nitrate ($\delta^{15}N\text{-}NO_3^-$, $\delta^{18}O\text{-}NO_3^-$) was assessed in order to elucidate ambient
$NO_x$ sources in two distinct areas of China. We also investigated the potential isotope
effect during the formation of nitrate aerosols from $NO_x$, and evaluated how disregard
of such N isotope fractionation can bias N-isotope mixing model-based estimates on
the $NO_x$ source apportionment for nitrate deposition.



## 2 Methods

### 2.1 Field sampling

In this study, PM$_{2.5}$ aerosol samples were collected on precombusted (450 °C for 6 hr) quartz filters (25 × 20 cm) on a day/night basis, using high-volume air samplers at a flow rate of 1.05 m$^3$ min$^{-1}$ in Sanjiang and Nanjing (Fig. 1). After sampling, the filters were wrapped in aluminum foil, packed in air-tight polyethylene bags and stored at -20 °C prior to further processing and analysis. Four blank filters were also collected. They were exposed for 10 min to ambient air (i.e., without active sampling). PM$_{2.5}$ mass concentration was analyzed gravimetrically (Sartorius MC5 electronic microbalance) with a ± 1 μg precision before and after sampling (at 25°C and 45 ± 5% during weighing).

**Figure 1.**

The Sanjiang campaign was performed during a period of intensive burning of agricultural residues between October 8 and 18, 2013, to examine if there is any significant difference between the $\delta^{15}$N values of $p$NO$_3^-$ and NO$_x$ emitted from biomass burning. The Sanjiang site (in the following abbreviated as SJ; 47.35°N, 133.31°E) is located at an ecological experimental station affiliated with the Chinese Academy of Sciences located in the Sanjiang Plain, a major agricultural area predominantly run by state farms in Northeastern China (Fig. 1). Surrounded by vast farm fields and bordering Far-Eastern Russia, SJ is situated in a remote and sparsely populated region, with a harsh climate and rather poorly industrialized economy. The annual mean temperature at SJ is close to the freezing point, with daily minima ranging between -31 and -15°C in the coldest month January. As a consequence of the relatively low temperatures (also during summer), biogenic production of NO$_x$ through soil microbial processes is rather



weak. SJ is therefore an excellent environment where to collect biomass burning-
emitted aerosols with only minor influence from other sources.
The Nanjing campaign was conducted between 17 December 2014 and 8 January 2015
with the main objective to examine whether N isotope measurements can be used as a
tool to elucidate $NO_x$ source contributions to ambient $pNO_3^-$ during times of severe
haze. Situated in the heartland of the lower Yangtze River region, Nanjing is, after
Shanghai, the second largest city in Eastern China. The aerosol sampler was placed at
the rooftop of a building on the Nanjing University of Information Science and
Technology campus (in the following abbreviated as NJ; 18 m a.g.l.; 32.21° N, 118.72°
E; Fig. 1), where $NO_x$ emissions derive from both industrial and transportation sources.
**2.2 Laboratory analysis**
The mass concentrations of inorganic ions (including $SO_4^{2-}$, $NO_3^-$, $Cl^-$, $NH_4^+$, $K^+$, $Ca^{2+}$,
$Mg^{2+}$, and $Na^+$), carbonaceous components (organic carbon or OC, elemental carbon or
EC), and water-soluble organic carbon or WSOC were determined using an ion
chromatograph (761 Compact IC, Metrohm, Switzerland), a thermal/optical OC/EC
analyzer (RT-4 model, Sunset Lab. Inc., USA), and a TOC analyzer (Shimadzu, TOC-
VCSH, Japan), respectively. Importantly, levoglucosan, a molecular marker for the
biomass combustion aerosols was detected using a Dionex$^{TM}$ ICS-5000$^+$ system
(Thermo Fisher Scientific, Sunnyvale, USA). In addition, a homologous series of
dicarboxylic acids ($C_2$-$C_{11}$) and related compounds (oxoacids, α-dicarbonyls and fatty
acids) were analyzed using an Agilent 7890 gas chromatography and GC-MS detection
(Agilent Technologies, Wilmington, USA), employing a dibutyl ester derivatization
technique. Chemical aerosol analyses, including sample pre-treatment, analytical
procedures, protocol adaption, detection limits, and experimental uncertainty were
described in detail in our previous work (Cao et al., 2016; Cao et al., 2017).
For isotopic analyses of aerosol nitrate, aerosol subsamples were generated by punching
1.4-cm disks out of the filters. In order to extract the $NO_3^-$, sample discs were placed in



acid-washed glass vials with 10 ml deionized water and placed in an ultra-sonic water
bath for 30 min. Between one and four disks were used for $NO_x$ extraction, dependent
on the aerosol $NO_3^-$ content on the filters, which was determined independently. The
extracts were then filtered (0.22 µm) and analyzed the next day. N and O isotope
analyses of the extracted/dissolved aerosol nitrate ($^{15}N/^{14}N$, $^{18}O/^{16}O$) were performed
using the denitrifier method (Sigman et al., 2001; Casciotti et al., 2002). Briefly, sample
$NO_3^-$ is converted to nitrous oxide ($N_2O$) by denitrifying bacteria that lack $N_2O$
reductase activity (*Pseudomonas chlororaphis* ATCC# 13985; formerly *Pseudomonas*
*aureofaciens*, referred to below as such). $N_2O$ is extracted, purified, and analyzed for
its N and O isotopic composition using a continuous-flow isotope ratio mass
spectrometer (Thermo Finnigan Delta$^+$, Bremen, German). Nitrate N and O isotope
ratios are reported in the conventional $\delta$-notation with respect to atmospheric $N_2$ and
standard mean ocean water (V-SMOW) respectively. Analyses are calibrated using the
international nitrate isotope standard IAEA-N3, with a $\delta^{15}N$ value of 4.7‰ and a $\delta^{18}O$
value of 25.6‰ (Böhlke et al., 2003). The blank contribution was generally lower than
0.2 nmol (as compared to 20 nmol of sample N). Based on replicate measurements of
standards and samples, the analytical precision for $\delta^{15}N$ and $\delta^{18}O$ was generally better
than ± 0.2‰ and ± 0.3‰ (1σ), respectively.
The denitrifier method generates $\delta^{15}N$ and $\delta^{18}O$ values of the combined pool of $NO_3^-$
and $NO_2^-$. The presence of substantial amounts of $NO_2^-$ in $NO_3^-$ samples may lead to
errors with regards to the analysis of $\delta^{18}O$ (Wankel et al., 2010). We refrained from
including a nitrite-removal step, because nitrite concentrations in our samples were
always < 1% of the $NO_3^-$ concentrations. In the following $\delta^{15}N_{NOx}$ and $\delta^{18}O_{NOx}$ are thus
referred to as nitrate $\delta^{15}N$ and $\delta^{18}O$ (or $\delta^{15}N_{NO3}$ and $\delta^{18}O_{NO3}$).
In the case of atmospheric/aerosol nitrate samples with comparatively high $\delta^{18}O$ values,
$\delta^{15}N$ values tend to be overestimated by 1-2‰ (Hastings et al., 2003), if the contribution
of $^{14}N^{14}N^{17}O$ to the $N_2O$ mass 45 signal is not accounted for during isotope ratio
analysis. For most natural samples, the mass-dependent relationship can be



approximated as $\delta^{17}O \approx 0.52 \times \delta^{18}O$, and the $\delta^{18}O$ can be used for the $^{17}O$ correction.
Atmospheric $NO_3^-$ does not follow this relationship but inhabits a mass-independent
component. Thus, we adopted a correction factor of 0.8 instead of 0.52 for the $^{17}O$ to
$^{18}O$ linearity (Hastings et al., 2003).

**2.3 Calculation of N isotope fractionation value ($\varepsilon_N$)**

As we described above, the transformation process of $NO_x$ to $HNO_3/NO_3^-$ involves
multiple reaction pathways (see also Fig. S1) and is likely to undergo isotope
equilibrium exchange reactions. The measured $\delta^{15}N\text{-}NO_3^-$ values of aerosol samples are
thus reflective of the combined N isotope signatures of various $NO_x$ sources ($\delta^{15}N\text{-}NO_x$)
plus any given N isotope fractionation. Recently, Walter and Michalski (2015) used a
computational quantum chemistry approach to calculate isotope exchange fractionation
factors for atmospherically relevant $NO_y$ molecules, and based on this approach, Zong
et al. (2017) estimated the N isotope fractionation during the transformation of $NO_x$ to
$pNO_3^-$ at a regional background site in China. Here we adopt, and slightly modify, the
approach by Walter and Michalski (2015) and Zong et al. (2017), and assumed that the
net N isotope effect $\varepsilon_N$ (for equilibrium processes A↔B: $\varepsilon_{A \leftrightarrow B} =$
$\left( \dfrac{(\text{heavy isotope/light isotope})_A}{(\text{heavy isotope/light isotope})_B} - 1 \right) \cdot 1000‰$ ; $\varepsilon_N$ refers to $\varepsilon_{N(NO_x \leftrightarrow pNO_3^-)}$ in this
study unless otherwise specified) during the gas-to-particle conversion from $NO_x$ to
$pNO_3^-$ formation ($\Delta(\delta^{15}N)_{pNO_3^- \text{-} NO_x} = \delta^{15}N\text{-}pNO_3^- - \delta^{15}N\text{-}NO_x \approx \varepsilon_N$) can be considered
a hybrid of the isotope effects of two dominant N isotopic exchange reactions:

$$\begin{aligned} \varepsilon_N &= \gamma \times \varepsilon_{N\left(NO_x \leftrightarrow pNO_3^-\right)_{OH}} + (1-\gamma) \times \varepsilon_{N\left(NO_x \leftrightarrow pNO_3^-\right)_{H_2O}} \\ &= \gamma \times \varepsilon_{N\left(NO_x \leftrightarrow HNO_3\right)_{OH}} + (1-\gamma) \times \varepsilon_{N\left(NO_x \leftrightarrow HNO_3\right)_{H_2O}} \end{aligned} \quad (1)$$

where $\gamma$ represents the contribution from isotope fractionation by the reaction of $NO_x$
and photo-chemically produced OH to form $HNO_3$ (and $pNO_3^-$), as shown by
$\varepsilon_{N\left(NO_x \leftrightarrow HNO_3\right)_{OH}}$ ($\varepsilon_{N\left(NO_x \leftrightarrow pNO_3^-\right)_{OH}}$). The remainder is formed by the hydrolysis of $N_2O_5$





with aerosol water to generate $HNO_3$ (and $pNO_3^-$), namely, $\varepsilon_{N(NO_x \leftrightarrow HNO_3)_{H_2O}}$
($\varepsilon_{N(NO_x \leftrightarrow pNO_3^-)_{H_2O}}$). Assuming that kinetic N isotope fractionation associated with the
reaction between $NO_x$ and OH is negligible, $\varepsilon_{N(NO_x \leftrightarrow pNO_3^-)_{OH}}$ can be calculated based on
mass-balance considerations:
$$\varepsilon_{N(NO_x \leftrightarrow pNO_3^-)_{OH}} = \varepsilon_{N(NO_x \leftrightarrow HNO_3)_{OH}} = \varepsilon_{N(NO_2 \leftrightarrow HNO_3)_{OH}}$$
$$= 1000 \times \left[ \frac{\left( {}^{15}\alpha_{NO_2/NO} - 1 \right)\left( 1 - f_{NO_2} \right)}{\left( 1 - f_{NO_2} \right) + \left( {}^{15}\alpha_{NO_2/NO} \times f_{NO_2} \right)} \right] \qquad (2)$$

where ${}^{15}\alpha_{NO_2/NO}$ is the temperature-dependent (see equation 7 and Table S1)
equilibrium N isotope fractionation factor between $NO_2$ and NO, and $f_{NO_2}$ is the
fraction of $NO_2$ in the total $NO_x$. $f_{NO_2}$ ranges from 0.2 to 0.95 (Walters and
Michalski, 2015). Similarly, assuming a negligible kinetic isotope fractionation
associated with the reaction $N_2O_5 + H_2O + aerosol \rightarrow 2HNO_3$, $\varepsilon_{N(NO_x \leftrightarrow pNO_3^-)_{H_2O}}$ can be
computed from the following equation:
$$\varepsilon_{N(NO_x \leftrightarrow pNO_3^-)_{H_2O}} = \varepsilon_{N(NO_x \leftrightarrow HNO_3)_{H_2O}} =$$
$$\varepsilon_{N(NO_x \leftrightarrow N_2O_5)_{H_2O}} = 1000 \times \left( {}^{15}\alpha_{N_2O_5/NO_2} - 1 \right) \qquad (3)$$

where ${}^{15}\alpha_{N_2O_5/NO_2}$ is the equilibrium isotope fractionation factor between $N_2O_5$ and
$NO_2$, which also is temperature-dependent (see equation 7 and Table S1).
Following Walter and Michalski (2015) and Zhong et al. (2017), $\gamma$ can then be
approximated based on the O isotope fractionation during the conversion of $NO_x$ to
$pNO_3^-$:
$$\varepsilon_{O(NO_x \leftrightarrow pNO_3^-)} = \gamma \times \varepsilon_{O(NO_x \leftrightarrow pNO_3^-)_{OH}} + \left( 1 - \gamma \right) \times \varepsilon_{O(NO_x \leftrightarrow pNO_3^-)_{H_2O}}$$
$$= \gamma \times \varepsilon_{O(NO_x \leftrightarrow HNO_3)_{OH}} + \left( 1 - \gamma \right) \times \varepsilon_{O(NO_x \leftrightarrow HNO_3)_{H_2O}} \qquad (4)$$



where $\varepsilon_{O\left(NO_x \leftrightarrow pNO_3^-\right)_{OH}}$ and $\varepsilon_{O\left(NO_x \leftrightarrow pNO_3^-\right)_{H_2O}}$ represent the O isotope effects associated
with $pNO_3^-$ generation through the reaction of $NO_x$ and OH to form $HNO_3$, and the
hydrolysis of $N_2O_5$ on a wetted surface to form $HNO_3$, respectively. $\varepsilon_{O\left(NO_x \leftrightarrow pNO_3^-\right)_{OH}}$ can
be further expressed as:

$$
\begin{aligned}
\varepsilon_{O\left(NO_x \leftrightarrow pNO_3^-\right)_{OH}} &= \varepsilon_{O\left(NO_x \leftrightarrow HNO_3\right)_{OH}} = \frac{2}{3}\varepsilon_{O\left(NO_2 \leftrightarrow HNO_3\right)_{OH}} + \frac{1}{3}\varepsilon_{O\left(NO \leftrightarrow HNO_3\right)_{OH}} \\
&= \frac{2}{3}\left[\frac{1000\left(^{18}\alpha_{NO_2/NO}-1\right)\left(1-f_{NO_2}\right)}{\left(1-f_{NO_2}\right)+\left(^{18}\alpha_{NO_2/NO} \times f_{NO_2}\right)} + \left(\delta^{18}O\text{-}NO_x\right)\right] + \\
&\quad \frac{1}{3}\left[\left(\delta^{18}O\text{-}H_2O\right) + 1000\left(^{18}\alpha_{OH/H_2O}-1\right)\right]
\end{aligned}
\tag{5}
$$

and $\varepsilon_{O\left(NO_x \leftrightarrow pNO_3^-\right)_{H_2O}}$ can be determined as follows:
$$\varepsilon_{O\left(NO_x \leftrightarrow pNO_3^-\right)_{H_2O}} = \varepsilon_{O\left(NO_x \leftrightarrow HNO_3\right)_{H_2O}} = \frac{5}{6}\left(\delta^{18}O\text{-}N_2O_5\right) + \frac{1}{6}\left(\delta^{18}O\text{-}H_2O\right) \tag{6}$$
where $^{18}\alpha_{NO_2/NO}$ and $^{18}\alpha_{OH/H_2O}$ represent the equilibrium O isotope fractionation
factors between $NO_2$ and NO, and OH and $H_2O$, respectively. The range of $\delta^{18}O\text{-}H_2O$
can be approximated using an estimated tropospheric water vapor $\delta^{18}O$ range of -25‰-
0‰. The $\delta^{18}O$ values for $NO_2$ and $N_2O_5$ range from 90‰ to 122‰ (Zong et al. 2017).
$^{15}\alpha_{NO_2/NO}$ and $^{15}\alpha_{N_2O_5/NO_2}$, $^{18}\alpha_{NO_2/NO}$ and $^{18}\alpha_{OH/H_2O}$ in these equations, are dependent
on the temperature, which can be expressed as:
$$1000\left(^m\alpha_{X/Y}-1\right) = \frac{A}{T^4} \times 10^{10} + \frac{B}{T^3} \times 10^8 + \frac{C}{T^2} \times 10^6 + \frac{D}{T} \times 10^4 \tag{7}$$
where A, B, C, and D are experimental constants (Table S1) over the temperature range
of 150-450 K (Walters and Michalski, 2015; Walters et al., 2016; Walters and Michalski,
2016; Zong et al., 2017).
Based on Equations 4-7 and measured values for $\delta^{18}O\text{-}pNO_3^-$ of ambient $PM_{2.5}$, a Monte





Carlo simulation was performed to generate 10000 feasible solutions. The error
between predicted and measured $\delta^{18}$O was less than 0.5‰. The range (maximum and
minimum) of computed contribution ratios ($\gamma$) were then integrated in Equation 1 to
generate an estimate range for the nitrogen isotope effect $\varepsilon_N$ (using Equations 2-3).
$\delta^{15}$N-$p$NO$_3^-$ values can be calculated based on $\varepsilon_N$ and the estimated $\delta^{15}$N range for
atmospheric NO$_x$, (see section 2.4).
**2.4 Bayesian isotope mixing model**
Isotopic mixing models allow estimating the relative contribution of multiple sources
(e.g., emission sources of NO$_x$) within a mixed pool (e.g., ambient $p$NO$_3^-$). By explicitly
considering the uncertainty associated with the isotopic signatures of any given source,
as well as isotope fractionation during the formation of various components of a mixture,
the application of Bayesian methods to stable isotope mixing models generates robust
probability estimates of source proportions, and are often more appropriate when
targeting natural systems than simple linear mixing models (Chang et al., 2016a). Here
the Bayesian model MixSIR (a stable isotope mixing model using sampling-
importance-resampling) was used to disentangle multiple NO$_x$ sources by generating
potential solutions of source apportionment as true probability distributions, which has
been widely applied in a number of fields. Details on the model frame and computing
methods are given in SI Text S1.
Here, coal combustion (13.72 ± 4.57‰), transportation (-3.71 ± 10.40‰), biomass
burning (1.04 ± 4.13‰), and biogenic emissions from soils (-33.77 ± 12.16‰) were
considered to be the most relevant contributors of NO$_x$ (Table S2 and Text S2). The
$\delta^{15}$N of atmospheric NO$_x$ is unknown. However, it can be assumed that its range in the
atmosphere is constrained by the $\delta^{15}$N of the NO$_x$ sources and the $\delta^{15}$N of $p$NO$_3^-$ after
equilibrium fractionation conditions have been reached. Following Zong et al. (2017),
$\delta^{15}$N-NO$_x$ in the atmosphere was determined performing iterative model simulations,
with a simulation step of 0.01 times the equilibrium fractionation value based on the
$\delta^{15}$N-NO$_x$ values of the emission sources (mean and standard deviation) and the





measured $\delta^{15}$N-$p$NO$_3^-$ of ambient PM$_{2.5}$ (Fig. S2).
**3 Results**
**3.1 Sanjiang in Northern China**
The $\delta^{15}$N-$p$NO$_3^-$ and $\delta^{18}$O-$p$NO$_3^-$ values of the eight samples collected from the
Sanjiang biomass burning field experiment, ranged from 9.54 to 13.77‰ (mean:
12.17‰) and 57.17 to 75.09‰ (mean: 63.57‰), respectively. In this study, atmospheric
concentrations of levoglucosan quantified from PM$_{2.5}$ samples collected near the sites
of biomass burning in Sanjiang vary between 4.0 and 20.5 µg m$^{-3}$, two to five orders of
magnitude higher than those measured during non-biomass burning season (Cao et al.,
2017; Cao et al., 2016). Levoglucosan is an anhydrosugar formed during pyrolysis of
cellulose at temperatures above 300 °C (Simoneit, 2002). Due to its specificity for
cellulose combustion, it has been widely used as a molecular tracer for biomass burning
(Simoneit et al., 1999; Liu et al., 2013a; Jedynska et al., 2014; Liu et al., 2014). Indeed,
the concentrations of levoglucosan and aerosol nitrate in Sanjiang were highly
correlated (R$^2$ = 0.64; Fig. 2a), providing compelling evidence that particulate nitrate
measured during our study period was predominately derived from biomass burning
emissions.
**3.2 Nanjing in Eastern China**
The mass concentrations $\left( mean_{min}^{max} \pm 1\sigma, n = 43 \right)$ of PM$_{2.5}$ and $p$NO$_3^-$ measured in Nanjing
City were $122.1_{39.0}^{227.8} \pm 47.9$ and $17.8_{4.0}^{45.2} \pm 10.3$ µg m$^{-3}$, respectively. All PM$_{2.5}$
concentrations exceeded the Chinese Air Quality Standard for daily PM$_{2.5}$ (35 µg m$^{-3}$),
suggesting severe haze pollution during the sampling period. The corresponding $\delta^{15}$N-
$p$NO$_3^-$ values (raw data without correction) ranged between 5.39‰ and 17.99‰,
indicating significant enrichment in $^{15}$N relative to rural and coastal marine atmospheric
NO$_3^-$ sources (Table S4). This may be due to the prominent contribution of fossil fuel-
related NO$_x$ emissions with higher $\delta^{15}$N values in urban areas (Elliott et al., 2007; Park





et al., 2018).

## 4 Discussion

### 4.1 Sanjiang campaign: theoretical calculation and field validation of N isotope fractionation during $p$NO$_3^-$ formation

To be used as a quantitative tracer of biomass-combustion-generated aerosols, levoglucosan must be conserved during its transport from its source, without partial removal by reactions in the atmosphere (Hennigan et al., 2010). The mass concentrations of non-sea-salt potassium (nss-K$^+$ = K$^+$ - 0.0355*Na$^+$) is considered as an independent/additional indicator of biomass burning (Fig. 2b). The association of elevated levels of levoglucosan with high nss-K$^+$ concentrations underscores that the two compounds derived from the same proximate sources, and that thus aerosol levoglucosan in Sanjiang was indeed pristine and represented a reliable source indicator that is unbiased by altering processes in the atmosphere. Moreover, in our previous work (Cao et al., 2017), we observed that there was a much greater enhancement of atmospheric NO$_3^-$ compared to SO$_4^{2-}$ (a typical coal-related pollutant). This additionally points to biomass burning, and not coal-combustion, as the dominant $p$NO$_3^-$ source in the study area, making SJ and ideal "quasi single source" environment for calibrating the N isotope effect during $p$NO$_3^-$ formation.

**Figure 2.**

Our $\delta^{18}$O-$p$NO$_3^-$ values are well within the broad range of values in previous reports (Zong et al., 2017; Geng et al., 2017; Walters and Michalski, 2016). However, as depicted in Fig. 3, the $\delta^{15}$N values of biomass burning-emitted NO$_3^-$ fall within the range of $\delta^{15}$N-NO$_x$ values typically reported for emissions from coal combustion, whereas they are significantly higher than the well-established values for $\delta^{15}$N-NO$_x$



emitted from the burning of various types of biomass (mean: 1.04 ± 4.13‰, ranging
from -7 to +12‰) (Fibiger and Hastings, 2016). Turekian et al. (1998) conducted
laboratory tests involving the burning of eucalyptus and African grasses, and
determined that the $\delta^{15}$N of $p$NO$_3^-$ (around 23‰) was 6.6‰ higher than the $\delta^{15}$N of the
burned biomass. This implies significant N isotope partitioning during biomass burning.
In the case of complete biomass combustion, by mass balance, the first gaseous
products (i.e., NO$_x$) have the same $\delta^{15}$N as the biomass. Hence any discrepancy between
the $p$NO$_3^-$ and the $\delta^{15}$N of the biomass can be attributed to the N isotope fractionation
associated with the partial conversion of gaseous NO$_x$ to aerosol NO$_3^-$. Based on the
computational quantum chemistry (CQC) module calculations, the N isotope
fractionation $\varepsilon_N$ $\left( mean_{min}^{max} \pm 1\sigma \right)$ determined from the Sanjiang data was
$10.99_{10.30}^{12.54} \pm 0.74\%_0$. After correcting the primary $\delta^{15}$N-$p$NO$_3^-$ values under the
consideration of $\varepsilon_N$, the resulting mean $\delta^{15}$N of $1.17_{-1.89}^{2.98} \pm 1.95\%_0$ is very close to the
N isotopic signature expected for biomass burning-emitted NO$_x$ (1.04 ± 4.13‰) (Fig.
3) (Fibiger and Hastings, 2016). The much higher $\delta^{15}$N-$p$NO$_3^-$ values in our study
compared to reported $\delta^{15}$N-NO$_x$ values for biomass burning can easily be reconciled
when including N isotope fractionation during the conversion of NO$_x$ to NO$_3^-$. Put
another way, given that Sanjiang is an environment where we can essentially exclude
NO$_x$ sources other than biomass burning at the time of sampling, the data nicely validate
our CQC module-based approach to estimate $\varepsilon_N$.

**Figure 3**.


**4.2 Source apportionment of NO$_x$ in an urban setting using a Bayesian isotopic**
**mixing model**
Due to its high population density and intensive industrial production, the Nanjing



atmosphere was expected to have high $NO_x$ concentrations derived from road traffic
and coal combustion (Zhao et al., 2015). However, the raw $\delta^{15}N$-$pNO_3^-$ values (10.93 ±
3.32‰) fell well within the variation range of coal-emitted $\delta^{15}N$-$NO_x$ (Fig. 3). It is
tempting to conclude that coal combustion is the main, or even sole, $pNO_3^-$ source
(given the equivalent $\delta^{15}N$ values), yet, this is very unlikely. The data rather confirm
that significant isotope fractionation occurred during the conversion of $NO_x$ to $NO_3^-$
and that, without consideration of the N isotope effect, traffic-related $NO_x$ emissions
will be markedly underestimated.
In the atmosphere, the oxygen atoms of $NO_x$ rapidly exchanged with $O_3$ in the $NO/NO_2$
cycle (see equations R1-R3) (Hastings et al., 2003), and the $\delta^{18}O$-$pNO_3^-$ values are
determined by its production pathways (R4-R7), rather than the sources of $NO_x$
(Hastings et al., 2003). Thus, $\delta^{18}O$-$pNO_3^-$ can be used to gain information on the
pathway of conversion of $NO_x$ to nitrate in the atmosphere (Fang et al., 2011). In the
computational quantum chemistry module used here to calculate isotope fractionation,
we assumed that two-thirds of the oxygen atoms in $NO_3^-$ derive from $O_3$ and one-third
from •OH in the •OH generation pathway (R4) (Hastings et al., 2003); correspondingly,
five sixths of the oxygen atoms then derived from $O_3$ and one sixth from •OH in the
$O_3/H_2O$ pathway (R5-R7). The assumed range for $\delta^{18}O$-$O_3$ and $\delta^{18}O$-$H_2O$ values were
90‰-122‰ and -25‰-0‰, respectively (Zong et al., 2017). The partitioning between
the two possible pathways was then assessed through Monte Carlo simulation (Zong et
al., 2017).   The estimated range was rather broad, given the wide range of $\delta^{18}O$-$O_3$ and
$\delta^{18}O$-$H_2O$ values used. Nevertheless, the theoretical calculation of the average
contribution ratio (γ) for nitrate formation in Nanjing via the reaction of $NO_2$ and •OH
is consistent with the results from simulations using the Weather Research and
Forecasting model coupled with Chemistry (WRF-Chem) (Fig. 4; see Text S3 for
details). A clear diurnal cycle of the mass concentration of nitrate formed through •OH
oxidation of $NO_2$ can be observed (Fig. S3), with much higher concentrations between
12:00 and 18:00, This indicates the importance of photochemically produced •OH
during daytime. Yet, throughout our sampling period in Nanjing, the average $pNO_3^-$



formation by the heterogeneous hydrolysis of $N_2O_5$ (12.6 µg m$^{-3}$) exceeded $p$NO$_3^-$
formation by the reaction of $NO_2$ and •OH (4.8 µg m$^{-3}$), even during daytime, consistent
with recent observations during peak pollution periods in Beijing (Wang et al., 2017).
Given that the production rates of $N_2O_5$ in the atmosphere is governed by ambient $O_3$
concentrations, reducing atmospheric $O_3$ levels appears to be one of the utmost
important measures to take for mitigating $p$NO$_3^-$ pollution in China's urban
atmospheres.

**Figure 4**.


In Nanjing, dependent on the time-dependent, dominant $p$NO$_3^-$ formation pathway, the
average N isotope fractionation value calculated using the computational quantum
chemistry module varied between 10.77‰ and 19.34‰ (15.33‰ on average). Using
the Bayesian model MixSIR, the contribution of each source can be estimated, based
on the mixed-source isotope data under the consideration of prior information on the
site (see Text S1 for detailed information regarding model frame and computing
method). As described above, theoretically, there are four major sources, i.e., road
traffic, coal combustion, biomass burning, and biogenic soil, potentially contributing to
ambient NO$_x$. As a start, we tentatively integrated all four sources into MixSIR (data
not shown). The relative contribution of biomass burning to the ambient NO$_x$ (median
value) ranged from 28% to 70% (average 42%), representing the most important source.
The primary reason for such apparently high contribution by biomass burning is that
the corrected $\delta^{15}$N-$p$NO$_3^-$ values of $-4.29^{0.42}_{-10.32} \pm 3.66$‰ are relatively close to the N
isotopic signature of biomass burning-emitted NO$_x$ (1.04 ± 4.13‰) compared to the
other possible sources. Based on $\delta^{15}$N alone, the isotope approach can be ambiguous if
there are more than two sources. The N isotope signature of NO$_x$ from biomass burning
falls right in between the spectrum of plausible values, with highest $\delta^{15}$N for emissions
from coal combustion on the one end, and much lower values for automotive and soil



emissions on the other, and will be similar to a mixed signature from coal combustion
and $NO_x$ emissions from traffic.
We can make several evidence-based pre-assumption to better constrain the emission
sources in the mixing model analysis: (1) sampling at a typical urban site in a major
industrial city in China, we can assume that the sources of road traffic and coal
combustion are dominant, while the contribution of biogenic soil to ambient $NO_x$
should have minimal impact, or can be largely neglected (Zhao et al., 2015); (2) there
is no crop harvest activity in Eastern China during the winter season. Furthermore,
deforestation and combustion of fuelwood has been discontinued in China's major
cities (Chang et al., 2016a). Therefore, the contribution of biomass burning-emitted
$NO_x$ during the sampling period should also be minor. Indeed, Fig. S4 shows that the
mass concentration of biomass burning-related $pNO_3^-$ is not correlated with the fraction
of levoglucosan that contributes to OC, confirming a weak impact of biomass burning
on the variation of $pNO_3^-$ concentration during our study period.
In a second, alternative, and more realistic scenario, we excluded biomass burning and
soil as potential source of $NO_x$ in MixSIR (see above). As illustrated in Fig. 5a,
assuming that $NO_x$ emissions in urban Nanjing during our study period originated
solely from road traffic and coal combustion, their relative contribution to the mass
concentration of $pNO_3^-$ is 12.5 ± 9.1 µg m$^{-3}$ (or 68 ± 11%) and 4.9 ± 2.5 µg m$^{-3}$ (or 32
± 11%), respectively. These numbers agree well with a city-scale $NO_x$ emission
inventory established for Nanjing recently (Zhao et al., 2015). Nevertheless, on a
nation-wide level, relatively large uncertainties with regards to the overall fossil fuel
consumption and fuel types propagate into large uncertainties of $NO_x$ concentration
estimates and predictions of longer-term emission trends (Li et al., 2017). Current
emission-inventory estimates (Jaegle et al., 2005; Zhang et al., 2012; Liu et al., 2015;
Zhao et al., 2013) suggest that in 2010 $NO_x$ emissions from coal-fired power plants in
China were about 30% higher than those from transportation. However, our isotope-
based source apportionment of $NO_x$ clearly shows that in 2014 the contribution from
road traffic to $NO_x$ emissions, at least in Nanjing (a city that can be considered



representative for most densely populated areas in China) is twice that of coal combustion. In fact, due to changing economic activities, emission sources of air pollutants in China are changing rapidly. For example, over the past several years, China has implemented an extended portfolio of plans to phase out its old-fashioned and small power plants, and to raise the standards for reducing industrial pollutant emissions (Chang, 2012). On the other hand, China continuously experienced double-digit annual growth in terms of auto sales during the 2000s, and in 2009 it became the world's largest automobile market (Liu et al., 2013b; Chang et al., 2017; Chang et al., 2016b). Recent satellite-based studies successfully analyzed the $NO_x$ vertical column concentration ratios for megacities in Eastern China and highlighted the importance of transportation-related NOx emissions (Reuter et al., 2014; Gu et al., 2014; Duncan et al., 2016; Jin et al., 2017). Moreover, long-term measurements of the ratio of $NO_3^-$ versus non-sea-salt $SO_4^{2-}$ in precipitation and aerosol jointly revealed a continuously increasing trend in Eastern China throughout the latest decade, suggesting decreasing emissions from coal combustion (Liu et al., 2013b; Itahashi et al., 2017). Both coal combustion- and road traffic-related $p$$NO_3^-$ concentrations are highly correlated with their corresponding tracers (i.e., $SO_2$ and $CO$, respectively), confirming the validity of our MixSIR modelling results. With justified confidence in our Bayesian isotopic model results, we conclude that previous estimates of $NO_x$ emissions from automotive/transportation sources in China based on bottom-up emission inventories may be too low.

**Figure 5**.

### 3.3 Previous $\delta^{15}N$-$NO_3^-$ based estimates on $NO_x$ sources

Stable nitrogen isotope ratios of nitrate have been used to identify nitrogen sources in various environments in China, often without large differences in $\delta^{15}N$ between



519 rainwater and aerosol $NO_3^-$ (Kojima et al., 2011). In previous work, no consideration

520 was given to potential N isotope fractionation during atmospheric $pNO_3^-$ formation.

521 Here, we reevaluated 700 data points of $\delta^{15}N$-$NO_3^-$ in aerosol (-0.77 ± 4.52‰, n = 308)

522 and rainwater (3.79 ± 6.14‰, n = 392) from 13 sites that are located in the area of

523 mainland China and the Yellow and East and South China Seas (Fig. 1), extracted from

524 the literature (see SI Table S4 for details). To verify the potentially biasing effects of

525 neglecting N isotope fractionation (i.e. testing the sensitivity of ambient $NO_x$ source

526 contribution estimates to the effect of N isotope fractionation), the Bayesian isotopic

527 mixing model was applied a) to the original $NO_3^-$ isotope data set and b) to the corrected

528 nitrate isotope data set, accounting for the N isotope fractionation during $NO_x$

529 transformation. All 13 sampling sites are located in non-urban areas; therefore, apart

530 from coal combustion and on-road traffic, the contributions of biomass burning and

531 biogenic soil to nitrate needs to be taken into account.

532 Although most of the sites are located in rural and coastal environments, using the

533 original data set without the consideration of N isotope fractionation in the Bayesian

534 isotopic mixing model, fossil fuel-related $NO_x$ emissions (coal combustion and on-road

535 traffic) appear as the largest contributor at all the sites (data are not shown). This is

536 particularly true for coal combustion: Everywhere, except for the sites of Dongshan

537 Islands and Mt. Lumin, $NO_x$ emissions seem to be dominated by coal combustion. Very

538 high contribution from coal combustion (on the order of 40-60%) particularly in

539 Northern China may be plausible, and can be attributed to a much larger consumption

540 of coal. Yet, rather unlikely, the highest estimated contribution of coal combustion (83%)

541 was calculated for Beihuang Island (a full-year sampling at a costal island that is 65 km

542 north of Shandong Peninsula and 185 km east of the Beijing-Tianjin-Hebei region) and

543 not for mainland China. While Beihuang may be an extreme example, we argue that,

544 collectively, the contribution of coal combustion to ambient $NO_x$ in China as calculated

545 on the basis of isotopic analyses in previous studies without the consideration of N

546 isotope fractionation represent overestimates.

547 As a first step towards a more realistic assessment of the actual partitioning of $NO_x$


sources in China in general (and coal combustion-emitted $NO_x$ in particular), it is
imperative to determine the location-specific values for $\varepsilon_N$. Unfortunately, without
$\delta^{18}O\text{-}NO_3^-$ data in hand, as well as data on meteorological parameters that correspond
to the 700 $\delta^{15}N\text{-}NO_3^-$ values used in our meta-analysis, it is not possible to estimate the
$\varepsilon_N$ values through the above-mentioned CQC module. As a viable alternative, we
adopted the approximate values for $\varepsilon_N$ as estimated in Sanjiang (10.99‰) and Nanjing
(15.33 ± 4.90‰). As indicated in Fig. 6, the estimates on the source partitioning is
sensitive to the choice of $\varepsilon_N$. Whereas with increasing $\varepsilon_N$, estimates on the relative
contribution of on-road traffic and biomass burning remained relatively stable;
estimates for coal combustion and biogenic soil changed significantly, in opposite
directions. More precisely, depending on $\varepsilon_N$, the average estimate of the fractional
contribution of coal combustion decreased drastically from 43% ($\varepsilon_N = 0$‰) to 5% ($\varepsilon_N$
$= 20$‰) (Fig. 6), while the contribution from biogenic soil to $NO_x$ emissions increased
in a complementary way. Given the lack of better constraints on $\varepsilon_N$ for the 13 sampling
sites, it cannot be our goal here to provide a robust revised estimate on the partitioning
of $NO_x$ sources throughout China and its neighboring areas. But we have very good
reasons to assume that disregard of N isotope fractionation during $pNO_3^-$ formation in
previous isotope-based source apportionment studies has likely led to overestimates of
the relative contribution of coal combustion to total $NO_x$ emissions in China. For what
we would consider the most conservative estimate, i.e. lowest calculated value for the
N isotope fractionation during the transformation of $NO_x$ to $pNO_3^-$ ($\varepsilon_N = 5$‰), the
approximate contribution from coal combustion to the $NO_x$ pool would be 28%, more
than 30% less than N isotope mixing model-based estimates would yield without
consideration of the N isotope fractionation (i.e., $\varepsilon N = 0$‰) (Fig. 6).

**Figure 6.**





## 4 Conclusion and outlook


Consistent with theoretical predictions, $\delta^{15}$N-$p$NO$_3^-$ data from a field experiment where
atmospheric $p$NO$_3^-$ formation could be attributed reliably to NO$_x$ from biomass burning
only, revealed that the conversion of NO$_x$ to $p$NO$_3^-$ is associated with a significant net
N isotope effect ($\varepsilon_N$). It is imperative that future studies, making use of isotope mixing
models to gain conclusive constraints on the source partitioning of atmospheric NO$_x$,
will consider this N isotope fractionation. The latter will change with time and space,
depending on the distribution of ozone and OH radicals in the atmosphere and the
predominant NO$_x$ chemistry. The O-isotope signatures of $p$NO$_3^-$ is mostly chemistry-
(and not source) driven (modulated by O-isotope exchange reactions in the atmosphere),
and thus, O isotope measurements do not allow addressing the ambiguities with regards
to the NO$_x$ source that may remain when just looking at $\delta^{15}$N values alone. However,
$\delta^{18}$O in $p$NO$_3^-$ will help assessing the relative importance of the dominant $p$NO$_3^-$
formation pathway. Simultaneous $\delta^{15}$N and $\delta^{18}$O measurements of atmospheric nitrate
thus allow reliable information on $\varepsilon_N$, and in turn on the relative importance of single
NO$_x$ sources. For example, for Nanjing, which can be considered representative for
other large cities in China, dual-isotopic and chemical-tracer evidence suggest that on-
road traffic and coal-fired power plants, rather than biomass burning, are the
predominant sources during high-haze pollution periods. Given that the increasing
frequency of nitrate-driven haze episodes in China, our findings are critically important
in terms of guiding the use of stable nitrate isotope measurements to evaluate the
relative importance of single NO$_x$ sources on regional scales, and for adapting suitable
mitigation measures. Future assessments of NO$_x$ emissions in China (and elsewhere)
should involve simultaneous $\delta^{15}$N and $\delta^{18}$O measurements of atmospheric nitrate and
NO$_x$ at high spatiotemporal resolution, allowing us to more quantitatively reevaluate
former N-isotope based NO$_x$ source partitioning estimates.
**Competing interests**




The authors declare that they have no competing interests.
**Data availability**
Data are available from the corresponding author on request.
**Acknowledgements**
This study was supported by the National Key Research and Development Program of
China (2017YFC0210101), the National Natural Science Foundation of China (Grant
nos. 91644103, 41705100, and 41575129), the Provincial Natural Science Foundation
of Jiangsu (BK20170946), the University Science Research Project of Jiangsu Province
(17KJB170011), and through University Basel Research funds.

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





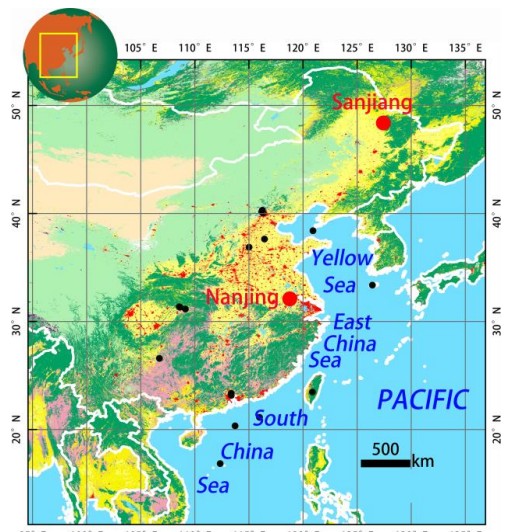


**Figure 1.** Location of the sampling sites Sanjiang and Nanjing. The black dots
indicate the location of sampling sites (sites are located in the area of mainland China
and the Yellow and East and South China Seas) with $\delta^{15}$N-NO$_3^-$ data from the
literature (see also Table S4).
















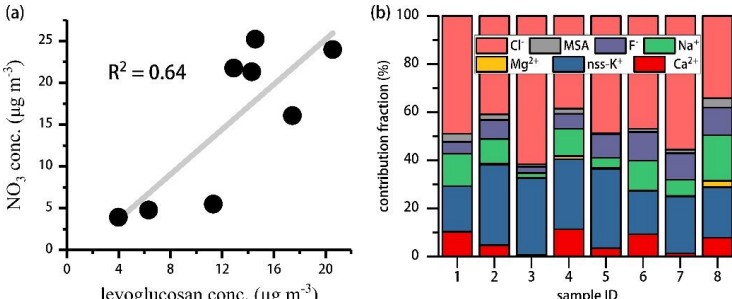

**Figure 2. (a)** Correlation analysis between the mass concentrations of levoglucosan and aerosol nitrate during the Sanjiang sampling campaign; **(b)** Variation of fractions of various inorganic species during day-night samplings at Sanjiang between 8 and October 2013 18 (sample ID 1 to 8, respectively). The higher relative abundances of nss-K$^+$ and Cl$^-$ are indicative for a biomass-burning dominated source. For sample ID information and exact sampling dates, refer to Table S3.



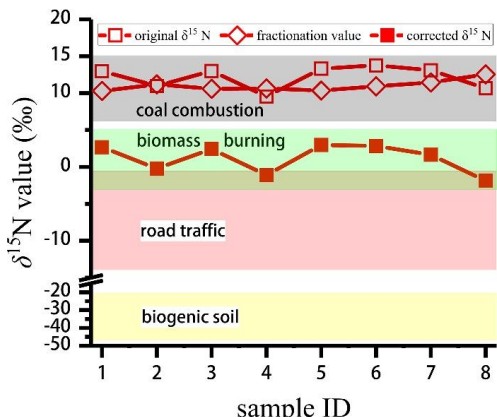

924

**Figure 3**. Original $\delta^{15}$N values ($\delta^{15}N_{ini}$) for $p$NO$_3^-$, calculated values for the N isotope

fractionation ($\varepsilon_N$) associated with the conversion of gaseous NO$_x$ to $p$NO$_3^-$, and

corrected $\delta^{15}$N values ($\delta^{15}N_{corr}$; $^{15}N_{ini}$ minus $\varepsilon_N$) of $p$NO$_3^-$ for each sample collected

during the Sanjiang sampling campaign. The colored bands represent the variation

range of $\delta^{15}$N values for different NO$_x$ sources based on reports from the literature

(Table S2). See Table S3 for the information regarding sample ID.







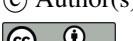

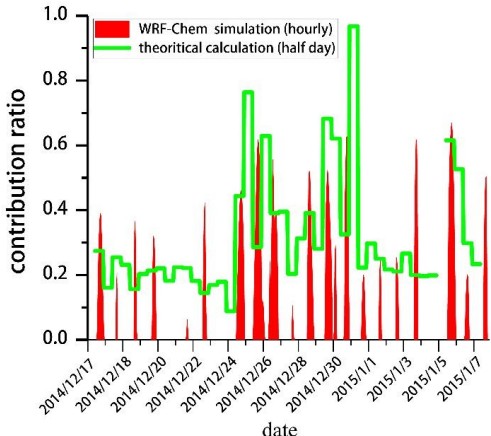


**Figure 4**. Comparison between the theoretical calculation and WRF-Chem simulation

of the average contribution ratio (γ) for nitrate formation in Nanjing via the reaction

of $NO_2$ and photochemically produced •OH.








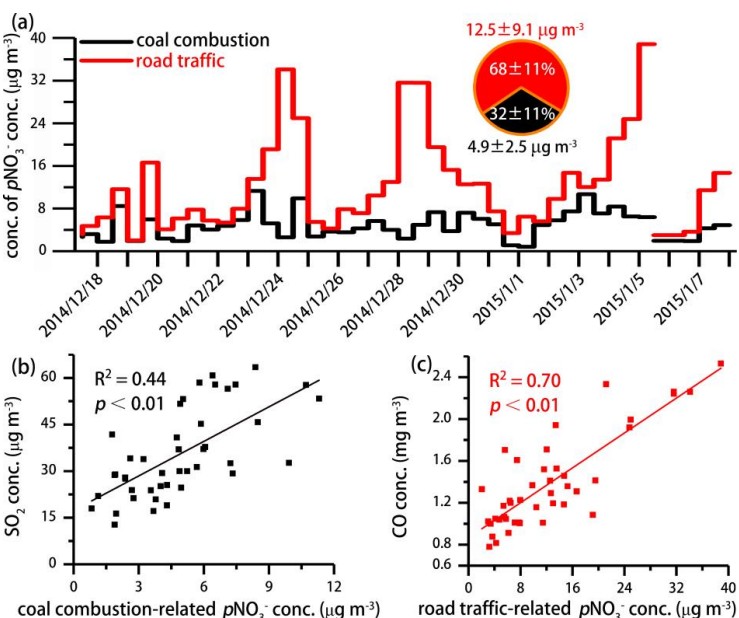

**Figure 5**. (**a**) Time-series variation of coal combustion and road traffic contribution to
the mass concentrations of ambient $p\mathrm{NO_3^-}$ in Nanjing, as estimated through MixSIR;
(**b**) Correlation analysis between the mass concentrations of coal combustion-related
$p\mathrm{NO_3^-}$ and $SO_2$; (**c**) Correlation analysis between the mass concentrations of road
traffic-related $p\mathrm{NO_3^-}$ and CO.



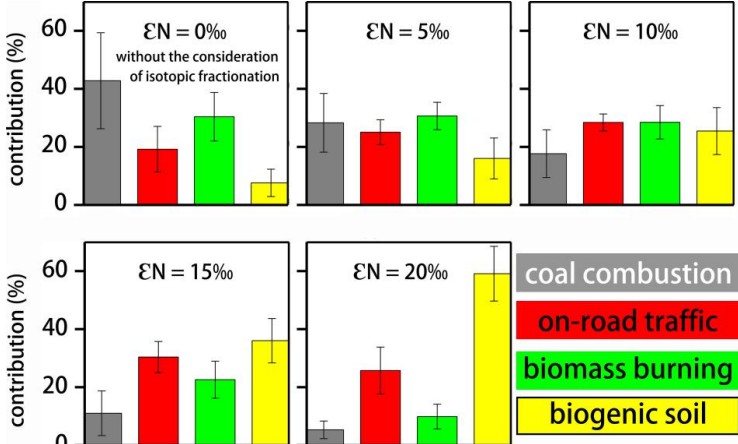


**Figure 6.** Estimates of the relative importance of single $NO_x$ sources (mean $\pm$ 1$\sigma$)

throughout China based on the original $\delta^{15}N$-$NO_3^-$ values extracted from the literature
($\varepsilon_N = 0‰$) and under consideration of significant N isotope fractionation during $NO_x$
transformation ($\varepsilon_N = 5‰$, 10‰, 15‰ or 20‰).