# Peer review of "Discussion started: 8 May 2018"

_Atmospheric Chemistry and Physics, 2018_

## Referee Comment (RC1) · Anonymous Referee #1 · 1 Jun 2018

Current application of stable isotopes in atmospheric particulate nitrate to partition NOx source contributions generally presupposes that nitrogen isotopic fractionation during the conversion of NOx to NO3- is minor. Here Chang et al. present a comprehensive evaluation of the nitrogen isotope fractionation during gas-to-particle conversion of NOx to NO3−. The computational quantum chemistry is applied to calculate the net N isotope effect (ÔŚN) associated with the conversion between NOx and NO3−, and validated through a source-specific monitoring campaign. The applicability of this method to atmospheric aerosol samples from a megacity shows satisfactory results,

which are in line to atmospheric chemistry modeling and to what one can expect in terms of source impact in a traffic-intensive environment. The source apportionment model to calculate nitrate fractions of different NOx sources is presented in a clear and concise way and is easily applicable by other researchers for similar studies. Great benefit with the method compared to other $\delta$15N-based source apportionment studies of atmospheric nitrate is the fact that coal combustion may be substantively overestimated in previous studies when the N isotope fractionation during atmospheric nitrate formation is neglected. This makes the study with more profound implications. I recommend this manuscript to be published in ACP with minor revision. 1. Title: replace "gas-particle" by "gas-to-particle" 2. L54: delete "nationwide," 3. L103: add relevant reference 4. L154-155: to my understanding, the source apportionment study of pNO3- was only performed in Nanjing 5. L190: it is a bit awkward to use "heartland" here 6. L203-207: I didn't find the data of dicarboxylic acids and related compounds. No need to mention the method here 7. L317: enough credits should be given to previous researchers 8. L319-321: although described in the SI, relevant references should be added 9. Figure 2b: what "MSA" stands for 10. Figure 6: replace "ÔŚN"
* * *

---

## Referee Comment (RC2) · Anonymous Referee #2 · 30 Jul 2018

Chang et al. propose a novel method to qualitatively determine the nitrogen isotope fractionation factor associated with NOx oxidation to form nitrate aerosols. The authors argue that the nitrogen isotope fractionation is a fundamentally important but overlooked factor in terms of influencing the source apportionment of particulate nitrate, particularly in urban polluted atmosphere. The explanations given are supported by strong observations, theory, and modeling. Overall, this work contributes a potentially powerful new tool for the investigation of atmospheric nitrate sources, and the isotopic fractionation that occurs during chemical processing. I have no major concerns regarding this manuscript. As mentioned by the first reviewer, it is well written, well presented and it makes sound. Beyond the remarks given by the first reviewer upon which I agree, I would appreciate if the authors can also consider the following points: (1) I assume that the authors have wrote a program that incorporated all of the equations in the MS to calculate the nitrogen isotope fractionation factor and estimate nitrate source attribution. I believe it will be a valuable asset if the authors could make such program publicly available; (2) compiled from previous studies, it is surprising to see no significant difference of $\delta$15N values among different phases of nitrate. How the authors explain my doubt; (3) the use of two pathways to explain the nitrogen isotope fractionation is classic and maybe correct to a large extent. I was wondering if other pathways to influence the nitrogen isotope fractionation and subsequently contribute to nitrate formation need to be mentioned at least; (4) The references in the Reference list are not always in the appropriate order: "Chang, Deng..., 2017" should come before "Chang, Liu..., 2016a". "Felix, J. D., and Elliott, E. M., 2014" should come before " Felix, J. D., Elliott, E. M., Gish, T. J...., 2013". "Felix, J. D., Elliott, E. M., and Shaw, S. L., 2012" should come after " Felix, J. D., Elliott, E. M., Gish, T. J...., 2013".

---

## Author Comment (AC1) · 31 Jul 2018

Our response to both reviewers' comments is uploaded as a *pdf file. Please see the attachment.

Please also note the supplement to this comment:
https://www.atmos-chem-phys-discuss.net/acp-2018-385/acp-2018-385-AC1-supplement.pdf

---

## Author Response (AR1)

We thank the reviewers for the valuable time and comments. Below, we respond to the reviewers' comments in detail and attach a marked-up manuscript (from page 8 to page

44) which highlights the changes made. Referee comments are in black, italic text. Our response to referees is in black, plain text.

**Referee #1**

Comment 1:

*Current application of stable isotopes in atmospheric particulate nitrate to partition*

*NOx source contributions generally presupposes that nitrogen isotopic fractionation*

*during the conversion of NOx to NO3- is minor. Here Chang et al. present a*

*comprehensive evaluation of the nitrogen isotope fractionation during gas-to-particle*

*conversion of NOx to NO3−. The computational quantum chemistry is applied to*

*calculate the net N isotope effect (ÔSN) associated with the conversion between NOx*

*and NO3´−, and validated through a source-specific monitoring campaign. The*

*applicability of this method to atmospheric aerosol samples from a megacity shows*

*satisfactory results, which are in line to atmospheric chemistry modeling and to what*

*one can expect in terms of source impact in a traffic-intensive environment. The source*

*apportionment model to calculate nitrate fractions of different NOx sources is presented*

*in a clear and concise way and is easily applicable by other researchers for similar*

*studies. Great benefit with the method compared to other δ15N-based source*

*apportionment studies of atmospheric nitrate is the fact that coal combustion may be*

*substantively overestimated in previous studies when the N isotope fractionation during*

*atmospheric nitrate formation is neglected. This makes the study with more profound*

*implications. I recommend this manuscript to be published in ACP with minor revision.*

Reply: We appreciate the reviewer for the recognition of this work, which give us a sense of accomplishment. Below please see our point-by-point reply.

Comment 2:

*Title: replace "gas-particle" by "gas-to-particle"*

Reply: We think that it is generally appropriate to use "gas-particle" here. In the revised

MS, we've replaced "gas-particle" by "gas-to-particle".

Comment 3:

*L54: delete "nationwide,"*

Reply: Revised accordingly.

Comment 4:

*L103: add relevant reference*

Reply: We added Morin et al. (2008) in the revised MS.

Morin, S., Savarino, J., Frey, M. M., Yan, N., Bekki, S., Bottenheim, J. W., and Martins,

J. M.: Tracing the origin and fate of $NO_x$ in the Arctic atmosphere using stable isotopes in nitrate, Science, 322, 730-732, doi: 10.1126/science.1161910, 2008.

Comment 5:

*L154-155: to my understanding, the source apportionment study of pNO3- was only*

*performed in Nanjing*

Reply: Sorry for our mistake. We change "in order to elucidate ambient $NO_x$ sources in two distinct areas of China" to "in order to elucidate ambient $NO_x$ sources in Nanjing

City of Eastern China".

Comment 6:

*L190: it is a bit awkward to use "heartland" here*

Reply: We deleted "the heartland of" in the revised MS.

Comment 7:

*L203-207: I didn't find the data of dicarboxylic acids and related compounds. No need*

*to mention the method here*

Reply: We deleted the description of this method in the revised MS.

Comment 8:

*L317: enough credits should be given to previous researchers. L319-321: although*

*described in the SI, relevant references should be added 9.*

Reply: Agree. Several relevant references have been added in the revised MS:

Parnell, A. C., Phillips, D. L., Bearhop, S., Semmens, B. X., Ward, E. J., Moore, J. W.,

Jackson, A. L., Grey, J., Kelly, D. J., and Inger, R.: Bayesian stable isotope mixing models, Environmetrics, 24, 387-399, doi: 10.1002/env.2221, 2013.

Phillips, D. L., Inger, R., Bearhop, S., Jackson, A. L., Moore, J. W., Parnell, A. C.,

Semmens, B. X., and Ward, E. J.: Best practices for use of stable isotope mixing models in food-web studies, Can. J. Zool., 92, 823-835, doi: 10.1139/cjz-2014-0127,

2014.

Zong, Z., Wang, X., Tian, C., Chen, Y., Fang, Y., Zhang, F., Li, C., Sun, J., Li, J., and

Zhang, G.: First assessment of $NO_x$ sources at a regional background site in North

China using isotopic analysis linked with modeling, Environ. Sci. Technol., 51, 5923-

5931, doi: 10.1021/acs.est.6b06316, 2017.

Comment 9:

*Figure 2b: what "MSA" stands for 10.*

Reply: "$MSA^-$" stands for "methyl sulphonate". We've added in the revised MS.

Comment 10

*Figure 6: replace "ÔSN"*

Reply: We guess "ÔSN" stands for "εN", and the reviewer want us to replace "εN" by

"$\varepsilon_N$". We revised Fig. 6 as follow:

[Figure]

**Referee #2**

Comment 1:

*Chang et al. propose a novel method to qualitatively determine the nitrogen isotope*

*fractionation factor associated with NOx oxidation to form nitrate aerosols. The*

*authors argue that the nitrogen isotope fractionation is a fundamentally important but*

*overlooked factor in terms of influencing the source apportionment of particulate*

*nitrate, particularly in urban polluted atmosphere. The explanations given are*

*supported by strong observations, theory, and modeling. Overall, this work contributes*

*a potentially powerful new tool for the investigation of atmospheric nitrate sources, and*

*the isotopic fractionation that occurs during chemical processing. I have no major*

*concerns regarding this manuscript. As mentioned by the first reviewer, it is well written,*

*well presented and it makes sound. Beyond the remarks given by the first reviewer upon*

*which I agree, I would appreciate if the authors can also consider the following points:*

Reply: We are thankful for the favorable comments. Below please see our point-by- point reply.

Comment 2:

*I assume that the authors have wrote a program that incorporated all of the equations*

*in the MS to calculate the nitrogen isotope fractionation factor and estimate nitrate*

*source attribution. I believe it will be a valuable asset if the authors could make such*

*program publicly available;*

Reply: This work was financially supported by the National Key Research and

Development Program of China, which require the submission of relevant software. We have the plan to make such program publicly available. However, we prefer not to publish the software at the present stage in order to avoid compromising the future of ongoing software registration. We are willing to share the software with the reviewer for reviewing purpose.

In "Data availability", we will remind readers to download the software through our group website (atmosgeochem.com) after the finish of software registration.

Comment 3:

*compiled from previous studies, it is surprising to see no significant difference of δ15N*

*values among different phases of nitrate. How the authors explain my doubt;*

Reply: We agree with the reviewer that different phases of nitrate generally have different variation range of $\delta^{15}N$ values. We only compiled the $\delta^{15}N$ data of particulate nitrate and precipitation nitrate from previous publications in this study. As a compromise, below we show the variation range of $\delta^{15}N$ values of $NH_x$ in all phases (paper in preparation). Firstly, gaseous $NO_x$ is as soluble as $NH_3$ in rainwater, and the ambient concentrations of HONO and $HNO_3$ are much lower than that of particulate nitrate. Thus, nitrate in precipitation is largely derived from particulate nitrate. In this regard, the difference of $\delta^{15}N$ values between particulate nitrate and precipitation nitrate can be expected to lower than the difference of $\delta^{15}N$ values between particulate ammonium and precipitation ammonium. Secondarily, in this study, we have no intention to accurately the determine the location-specific values for $\varepsilon_N$ in previous studies. Instead, the $\varepsilon_N$ was assigned by a large range of $\delta^{15}N$ values (from 0‰ to 20‰), which could significantly diminish the potential effects of the $\delta^{15}N$ gap between particulate nitrate and precipitation nitrate on the results of nitrate source apportionment.

[Figure]

Comment 4:

*the use of two pathways to explain the nitrogen isotope fractionation is classic and*

*maybe correct to a large extent. I was wondering if other pathways to influence the*

*nitrogen isotope fractionation and subsequently contribute to nitrate formation need to*

*be mentioned at least*;

Reply: We've enriched the discussion regarding the pathways of nitrate formation in the introduction section. Indeed, the co-editor also pointed out that the direct reactive uptake of $NO_3$ radicals by aerosol particles also contribute to particulate nitrate. Knopf et al. (2006, 2011) and Shiraiwa et al. (2012) have shown that $NO_3$ can be taken up efficiently by organic (e.g., levoglucosan) aerosol and may dominate oxidation of aerosol in the polluted urban nighttime (Kaiser et al., 2011). Globally, theoretical modeling results show that nearly 76%, 18%, and 4% of annual inorganic nitrate are formed via pathways/reactions involving OH, $N_2O_5$, and DMS or HC ($NO_3$ reacts with dimethylsulfide (DMS) or hydrocarbons (HC) predominantly at night) (e.g., Alexander et al., 2009). The stable O isotopic composition of atmospheric nitrate is a powerful proxy for assessing which oxidation pathways are important for converting $NO_x$ into nitrate under changing environmental conditions (e.g., polluted, volcanic events, climate change). In the same line, in this study, the average $\delta^{18}O$ value of $p$NO$_3^-$ in

Nanjing City was $83.0 \pm 11.2$‰ (see discussion later), suggesting that $p$NO$_3^-$ formation is dominated by the pathways of "OH + $NO_2$" and the heterogeneous hydrolysis of

$N_2O_5$.

Reply: Revised accordingly.

[revised manuscript text omitted]

$p$NO$_3^-$ formation ($\Delta\left(\delta^{15}\mathrm{N}\right)_{p\mathrm{NO}_3^- \text{-} \mathrm{NO}_x} = \delta^{15}\mathrm{N}\text{-}p\mathrm{NO}_3^- - \delta^{15}\mathrm{N}\text{-}\mathrm{NO}_x \approx \varepsilon_\mathrm{N}$) can be considered a hybrid of the isotope effects of two dominant N isotopic exchange reactions:

$$
\begin{aligned}
\varepsilon_\mathrm{N} &= \gamma \times \varepsilon_{\mathrm{N}\left(\mathrm{NO}_x \leftrightarrow p\mathrm{NO}_3^-\right)_{\mathrm{OH}}} + \left(1-\gamma\right) \times \varepsilon_{\mathrm{N}\left(\mathrm{NO}_x \leftrightarrow p\mathrm{NO}_3^-\right)_{\mathrm{H}_2\mathrm{O}}} \\
&= \gamma \times \varepsilon_{\mathrm{N}\left(\mathrm{NO}_x \leftrightarrow \mathrm{HNO}_3\right)_{\mathrm{OH}}} + \left(1-\gamma\right) \times \varepsilon_{\mathrm{N}\left(\mathrm{NO}_x \leftrightarrow \mathrm{HNO}_3\right)_{\mathrm{H}_2\mathrm{O}}}
\end{aligned} \tag{1}
$$

where $\gamma$ represents the contribution from isotope fractionation by the reaction of NO$_x$

and photo-chemically produced OH to form HNO$_3$ (and $p$NO$_3^-$), as shown by

$\varepsilon_{\mathrm{N}\left(\mathrm{NO}_x \leftrightarrow \mathrm{HNO}_3\right)_{\mathrm{OH}}}$ ($\varepsilon_{\mathrm{N}\left(\mathrm{NO}_x \leftrightarrow p\mathrm{NO}_3^-\right)_{\mathrm{OH}}}$). The remainder is formed by the hydrolysis of N$_2$O$_5$

with aerosol water to generate HNO$_3$ (and $p$NO$_3^-$), namely, $\varepsilon_{\mathrm{N}\left(\mathrm{NO}_x \leftrightarrow \mathrm{HNO}_3\right)_{\mathrm{H}_2\mathrm{O}}}$

($\varepsilon_{\mathrm{N}\left(\mathrm{NO}_x \leftrightarrow p\mathrm{NO}_3^-\right)_{\mathrm{H}_2\mathrm{O}}}$). Assuming that kinetic N isotope fractionation associated with the reaction between NO$_x$ and OH is negligible, $\varepsilon_{\mathrm{N}\left(\mathrm{NO}_x \leftrightarrow p\mathrm{NO}_3^-\right)_{\mathrm{OH}}}$ can be calculated based on mass-balance considerations:

$$
\begin{aligned}
\varepsilon_{\mathrm{N}\left(\mathrm{NO}_x \leftrightarrow p\mathrm{NO}_3^-\right)_{\mathrm{OH}}} &= \varepsilon_{\mathrm{N}\left(\mathrm{NO}_x \leftrightarrow \mathrm{HNO}_3\right)_{\mathrm{OH}}} = \varepsilon_{\mathrm{N}\left(\mathrm{NO}_2 \leftrightarrow \mathrm{HNO}_3\right)_{\mathrm{OH}}} \\
&= 1000 \times \left[ \frac{\left(^{15}\alpha_{\mathrm{NO}_2/\mathrm{NO}} - 1\right)\left(1 - f_{\mathrm{NO}_2}\right)}{\left(1 - f_{\mathrm{NO}_2}\right) + \left(^{15}\alpha_{\mathrm{NO}_2/\mathrm{NO}} \times f_{\mathrm{NO}_2}\right)} \right]
\end{aligned} \tag{2}
$$

where $^{15}\alpha_{\mathrm{NO}_2/\mathrm{NO}}$ is the temperature-dependent (see equation 7 and Table S1)

equilibrium N isotope fractionation factor between NO$_2$ and NO, and $f_{\mathrm{NO}_2}$ is the fraction of NO$_2$ in the total NO$_x$. $f_{\mathrm{NO}_2}$ ranges from 0.2 to 0.95 (Walters and

Michalski, 2015). Similarly, assuming a negligible kinetic isotope fractionation associated with the reaction N$_2$O$_5$ + H$_2$O + aerosol $\rightarrow$ 2HNO$_3$, $\varepsilon_{\mathrm{N}\left(\mathrm{NO}_x \leftrightarrow p\mathrm{NO}_3^-\right)_{\mathrm{H}_2\mathrm{O}}}$ can be computed from the following equation:

$$
\begin{aligned}
\varepsilon_{\mathrm{N}\left(\mathrm{NO}_x \leftrightarrow p\mathrm{NO}_3^-\right)_{\mathrm{H}_2\mathrm{O}}} &= \varepsilon_{\mathrm{N}\left(\mathrm{NO}_x \leftrightarrow \mathrm{HNO}_3\right)_{\mathrm{H}_2\mathrm{O}}} = \\
\varepsilon_{\mathrm{N}\left(\mathrm{NO}_x \leftrightarrow \mathrm{N}_2\mathrm{O}_5\right)_{\mathrm{H}_2\mathrm{O}}} &= 1000 \times \left(^{15}\alpha_{\mathrm{N}_2\mathrm{O}_5/\mathrm{NO}_2} - 1\right)
\end{aligned} \tag{3}
$$

[revised manuscript text omitted]